# Remote homology search with hidden Potts models

**Grey W. Wilburn****[1], Sean R. Eddy**[2,3]*

**1** Department of Physics, Harvard University, Cambridge, Massachusetts, United States of America,
**2** Howard Hughes Medical Institute, Department of Molecular and Cellular Biology, Harvard University,
Cambridge, Massachusetts, United States of America, **3** John A Paulson School of Engineering and Applied
Sciences, Harvard University, Cambridge, Massachusetts, United States of America

* seaneddy@fas.harvard.edu

## Abstract

Most methods for biological sequence homology search and alignment work with primary
sequence alone, neglecting higher-order correlations. Recently, statistical physics models
called Potts models have been used to infer all-by-all pairwise correlations between sites
in deep multiple sequence alignments, and these pairwise couplings have improved 3D
structure predictions. Here we extend the use of Potts models from structure prediction to
sequence alignment and homology search by developing what we call a hidden Potts
model (HPM) that merges a Potts emission process to a generative probability model of
insertion and deletion. Because an HPM is incompatible with efficient dynamic program-
ming alignment algorithms, we develop an approximate algorithm based on importance
sampling, using simpler probabilistic models as proposal distributions. We test an HPM
implementation on RNA structure homology search benchmarks, where we can compare
directly to exact alignment methods that capture nested RNA base-pairing correlations
(stochastic context-free grammars). HPMs perform promisingly in these proof of principle
experiments.

homology search with hidden Potts models. PLoS
Comput Biol 16(11): e1008085. https://doi.org/

Madison, UNITED STATES

**Data Availability Statement:** All relevant data are
within the manuscript and its Supporting
information files. Additionally, all relevant data will
be posted publicly without limitations on http://
eddylab.org/publications.html.

## Author summary

Computational homology search and alignment tools are used to infer the functions and
evolutionary histories of biological sequences. Most widely used tools for sequence
homology searches, such as BLAST and HMMER, rely on primary sequence conservation
alone. It should be possible to make more powerful search tools by also considering
higher-order covariation patterns induced by 3D structure conservation. Recent advances
in 3D protein structure prediction have used a class of statistical physics models called
Potts models to infer pairwise correlation structure in multiple sequence alignments.
However, Potts models assume alignments are given and cannot build new alignments,
limiting their use in homology search. We have extended Potts models to include a proba-
bility model of insertion and deletion so they can be applied to sequence alignment and
remote homology search using a new model we call a hidden Potts model (HPM). Tests of

**Funding:** SRE and GWW are funded by the Howard Hughes Medical Institute. (https://www.hhmi.org/). SRE is a Howard Hughes Medical Institute Investigator (grant number n/a). SRE is funded by National Human Genome Research Institute of the National Institutes of Health award R01-HG009116 (https://www.genome.gov/). Some ideas in this work were conceived at workshops hosted at the Centro de Ciencias de Benasque Pedro Pascual in Benasque, Spain (http://www.benasque.org/), and the Aspen Center for Physics, supported by National Science Foundation grant PHY-1066293 (https://www.nsf.gov/). The content is solely the responsibility of the authors and does not necessarily represent the official views of the National Institutes of Health. The funders had no role in study design, data collection and analysis, decision to publish, or preparation of the manuscript.

**Competing interests:** The authors have declared that no competing interests exist.

our prototype HPM software show promising results in initial benchmarking experiments, though more work will be needed to use HPMs in practical tools.

## Introduction

An important task in bioinformatics is determining whether a new sequence of unknown biological function is evolutionarily related, or homologous, to other known sequences or families of sequences. Critical to the concept of homology is alignment: homology tools create multiple sequence alignments (MSAs) in which evolutionarily related positions are aligned in columns by inferring patterns of sequence conservation induced by complex evolutionary constraints maintaining the structure and function of the sequence [1].

Homology search tools are used in a wide range of biological problems, but it is common for these tools to fail to identify distantly related sequences; many genes evolve quickly enough that homologs may exist yet be undetectable [2]. More powerful and sensitive homology search tools are therefore needed.

One possible way to improve the sensitivity of homology search and alignment is to develop new methods that successfully capture patterns of residue correlation induced by 3D structural constraints. State-of-the-art homology search methods do not model certain important elements of structure-induced conservation. Methods such as BLAST and HMMER, the latter of which uses profile hidden Markov models (pHMMs), align and score sequences using primary sequence conservation alone [3–5]. Specific methods for RNA homology search, such as the software package Infernal, use a class of probabilistic models called profile stochastic context free grammars (pSCFGs) to infer primary *and* secondary structure conservation in RNA [6, 7]. However, Infernal is limited to nested, disjoint pairs of nucleotides, meaning it cannot capture complicated 3D RNA structural elements like pseudoknots and base triples, let alone complex correlation structure in protein MSAs.

An opportunity to build a more sensitive sequence homology search method has arisen via two key developments. First, there has been recent exciting progress in exploring pairwise correlations between columns in protein and RNA MSAs using Potts model (aka Markov random field) methods from statistical physics [8–14]. Previous work has applied Potts models to homology search and alignment problems with specific proteins, but not to biological sequences in general [15–21]. Potts models have also been used to study protein-protein interactions [22–25], mutational effects [26–30], cellular morphogenesis [31], and collective neuron function [32]. Building upon previous methods that use pairwise sequence correlation to infer conserved base pairs in RNA structure and 3D structure in proteins [33, 34], a Potts model expresses the probability that a particular sequence belongs to a family represented by an MSA as a function of all possible characters (amino acids or nucleotides) at each position and all possible pairs of characters across all positions.

Second, very large datasets of aligned homologous protein and RNA sequences are available, such as Pfam for protein and Rfam for RNA [35, 36]. Deep MSAs allow for new methods that analyze more subtle patterns of sequence conservation.

Potts models have two drawbacks for applications in sequence homology search. First, Potts models assume *fixed length* sequences in *given* MSAs, treating insertions and deletions as an extra character (5th nucleotide or 21st amino acid). Homology search involves scoring *variable length* sequences and *inferring* alignments. Second, models such as pHMMs and pSCFGs use efficient dynamic programming algorithms that align and score sequences in polynomial

time. Unfortunately, the all-by-all pairwise nature of Potts models makes them incompatible with dynamic programming algorithms and therefore impractical for homology search.

It would be useful to have a homology search tool that can both handle higher-order sequence correlation patterns, as a Potts model can, *and* efficiently align and score variable length sequences, as BLAST, HMMER, and Infernal are able to do. Inspired by early work using Potts models for protein alignment and the success of Potts models in structure prediction, we have developed a class of models called hidden Potts models (HPMs) that combines the coupled Potts emission process with a probabilistic insert-deletion model. In addition, we have developed an approximate algorithm that uses importance sampling to align and score variable length sequences to a parameterized HPM.

We have built a proof of principle HPM software implementation. To test the efficacy of HPMs in homology search and alignment, we compare our software to HMMER and Infernal in RNA remote homology benchmark tests based on trusted, hand-curated non-coding RNA alignments. RNA is an ideal testbed, as secondary and tertiary structure are largely dictated by stereotyped base-pairing interactions, making model parameterization interpretable. Also, pSCFGs capture nested pairwise correlations, a key part of RNA homology search, leading to a natural comparison to an existing approach that captures much of the pairwise residue correlation structure in conserved RNAs. As HPMs capture nested *and* non-nested pairwise correlations, we expect them to at least equal, if not outperform, pSCFGs in RNA remote homology search and alignment.

## Results

### Hidden Potts models emit variable length sequences

A generative homology model describes the probability of creating an unaligned sequence of $L$ observed characters, $\vec{x}$. In the case of biological sequences, characters in $\vec{x}$ represent monomer units in a specific alphabet (20 amino acids for protein, 4 nucleotides for DNA and RNA).

A Potts model expresses the probability of a homologous sequence as a function of primary conservation *and* all possible pairwise correlations between all consensus sites in a biological sequence (i.e., consensus columns in a multiple sequence alignment, where we define consensus columns as those having fewer than 50% gap characters). Going beyond a Potts model, a hidden Potts model consists of $M$ sites called *match states* (squares in Fig 1), corresponding to well-conserved columns in an MSA, and intermediate insert states (diamonds) that handle variable-length insertions between match states. An HPM has a fixed number of match states, which restricts the model to *glocal* alignment (local to the sequence but global to the model).

An HPM is a hybrid between a profile hidden Markov model (pHMM) [4] and a Potts model. Much like a pHMM, an HPM consists of an observed sequence $\vec{x}$ and a *hidden* state path $\vec{\sigma}$ representing the alignment of $\vec{x}$ to the model. $\vec{\sigma}$ is a Markov chain, with the individual states mapping to columns in a multiple sequence alignment. Each state *emits* an observed character, with all emissions combining to form $\vec{x}$.

Unlike a pHMM, HPM states do not emit characters independent of other states, and deletions relative to the model's consensus are handled differently. HPM match states generate a set of characters with a single, correlated emission from a Potts model. As Potts models can only handle fixed length sequences, HPMs do not model the lack of a character in a consensus column with a special delete state in $\vec{\sigma}$, but rather as the emission of a "dummy" deletion character from a match state. The deletion character effectively serves as an extra character in the alphabet (21st amino acid or 5th nucleotide). As such, we divide $\vec{x}$ into two non-contiguous subsequences: $\vec{x}_m$, the characters emitted from match states (which may include deletion characters); and $\vec{x}_i$, the characters generated by insert states (all of which represent monomers). In

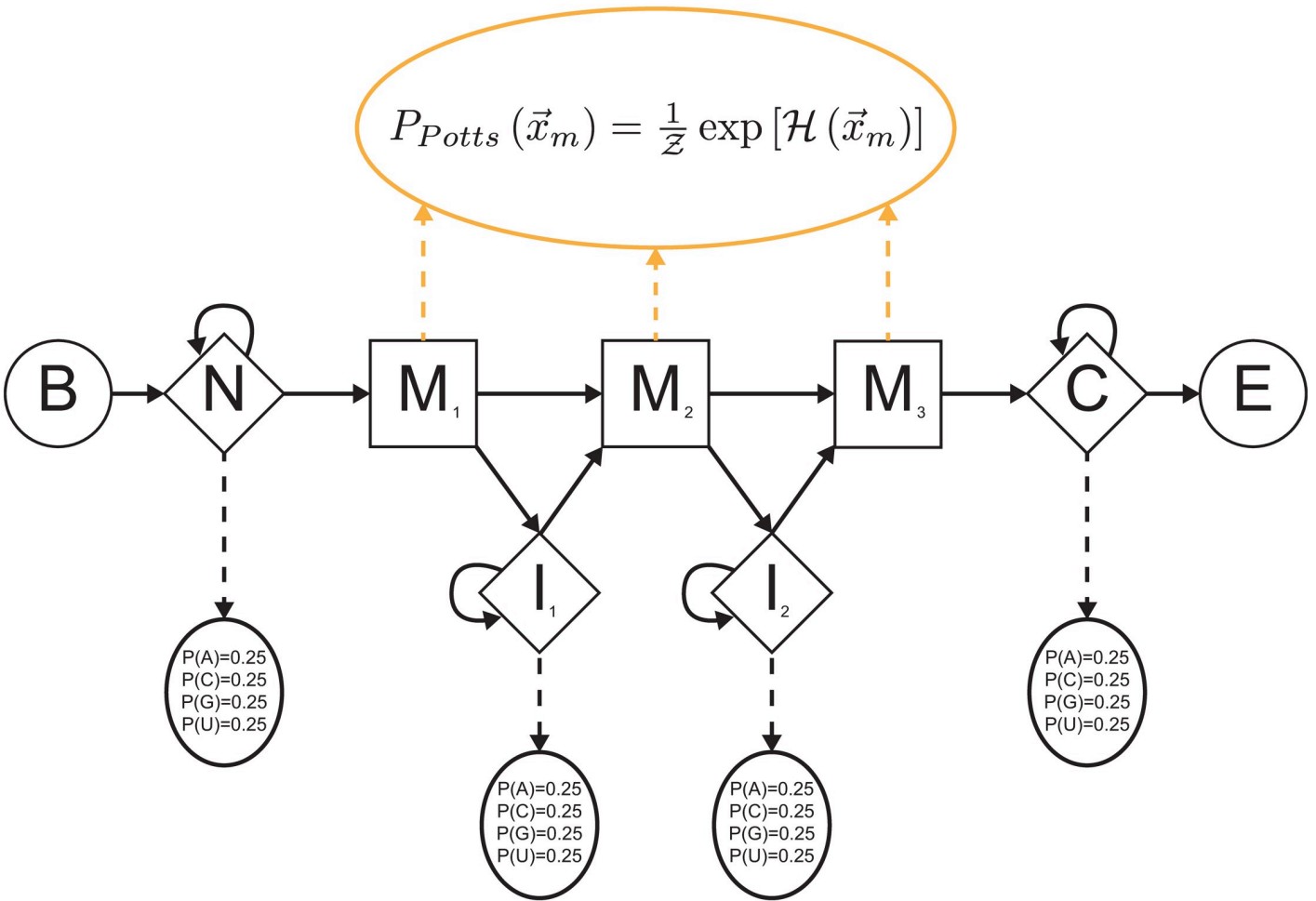

**Fig 1. Hidden Potts model architecture.** Squares are conserved match states and diamonds are insert states. No delete states exist. Silent begin and end states are represented by circles. An HPM is a hybrid between a Potts model and a pHMM: correlated character generation (including deletion "characters" rather than delete states) in match columns (consensus sites in an MSA) comes from a Potts distribution (dotted orange arrows), while transition probabilities linking states and site-independent character emissions in unaligned insert columns come from a pHMM (solid and dashed black arrows, respectively).

this notation, *m* signifies "match" and *i* represents "insert". A pHMM path, which we represent as $\vec{\pi}$ and can include delete states, is not identical to a corresponding HPM path: a single HPM path $\vec{\sigma}$ can map to up to $2^M$ possible paths under a pHMM, representing the possible distribution of deletions in consensus columns.

Mathematically, $\vec{\sigma}$, $\vec{x}_m$, and $\vec{x}_i$ are nuisance variables that must be marginalized over to obtain $P(\vec{x})$. The joint probability of $\vec{x}$, $\vec{x}_m$, $\vec{x}_i$, and $\vec{\sigma}$ under an HPM can be factorized into two terms.

$$P\left(\vec{x}, \vec{x}_m, \vec{x}_i, \vec{\sigma}\right) = P\left(\vec{x}|\vec{x}_m, \vec{x}_i, \vec{\sigma}\right) P\left(\vec{x}_m, \vec{x}_i, \vec{\sigma}\right) \tag{1}$$

Given $\vec{\sigma}$, $\vec{x}_m$ and $\vec{x}_i$ can be dealigned to produce a unique, unaligned sequence $\vec{x}$; $P(\vec{x}|\vec{x}_m, \vec{x}_i, \vec{\sigma})$ is 1 if $\vec{x}_m$, $\vec{x}_i$, and $\vec{\sigma}$ can be combined to produce unaligned sequence $\vec{x}$ and 0 otherwise.

Each of $\vec{x}_m$, $\vec{x}_i$, and $\vec{\sigma}$ is generated differently under an HPM. Their joint probability can be factored into three terms: $P_t(\vec{\sigma})$, a product of transition probabilities linking states to one another (solid black arrows in Fig 1); $P_{Potts}(\vec{x}_m)$, describing the Potts emission of characters from match states (dashed orange arrows); and $P_i(\vec{x}_i|\vec{\sigma})$, a product of independent emission of

characters from insert states (dashed black arrows). In this notation, $t$ is shorthand for "transition."

$$P\left(\vec{x}_m, \vec{x}_i, \vec{\sigma}\right) = P_{Potts}\left(\vec{x}_m\right) P_i\left(\vec{x}_i | \vec{\sigma}\right) P_t\left(\vec{\sigma}\right) \tag{2}$$

The entire match sequence $\vec{x}_m$ is generated by one multi-character emission from the Potts distribution. The Potts distribution assumes the probability of generating a sequence from match states is given by an exponential Boltzmann distribution:

$$P_{Potts}(\vec{x}_m) = \frac{1}{Z} e^{\mathcal{H}(\vec{x}_m)} \tag{3}$$

Here $\mathcal{H}(\vec{x}_m)$ is referred to as the *Hamiltonian*, while $Z$ is the *partition function*, a normalization constant. The Hamiltonian is given by:

$$\mathcal{H}(\vec{x}_m) = \sum_{k=1}^{M} h_k(x_{m_k}) + \sum_{k=1}^{M}\sum_{l=k+1}^{M} e_{kl}(x_{m_k}, x_{m_l}) \tag{4}$$

$h_k(x_{m_k})$ represents the *single position preference* for character $x_{m_k}$ in match state $m_k$. $e_{kl}(x_{m_k}, x_{m_l})$ corresponds to the *statistical coupling* between character preferences at separate match columns $k$ and $l$. These are the Potts model parameters that we estimate from an input MSA.

The insert emission probability factorizes into independent terms for each character in $\vec{x}_i$.

$$P_i(\vec{x}_i | \vec{\sigma}) = \prod_{j=1}^{I} P\left(x_{i_j}\right) \tag{5}$$

The probability of a state path $\vec{\sigma}$ is given by:

$$P_t(\vec{\sigma}) = P\left(\sigma_1\right) \prod_{n=2}^{\Lambda} P\left(\sigma_n | \sigma_{n-1}\right) \tag{6}$$

Here, $\sigma_n$ is an HPM state. $\Lambda$ is the total number of states in $\vec{\sigma}$. Given that $\vec{x}_m$ can include deletion characters, $\Lambda$ is at least as large as $L$, the total number of non-gap characters observed in sequence $\vec{x}$.

HPMs are generative models capable of producing any sequence, while the inclusion of possible self-transitions within insert states allows for the generation of sequences of any length. The steps to generate a sequence are:

1. Choose the path $\vec{\sigma}$ by performing a random walk along the state path following $P_t(\vec{\sigma})$.

2. Choose insert state characters by independently drawing from $P_i(\vec{x}_i | \vec{\sigma})$.

3. Generate match column characters from a single emission from $P_{Potts}(\vec{x}_m)$.

4. Dealign the sequence by combining $\vec{x}_m$ and $\vec{x}_i$ and removing deletion characters. This corresponds to choosing the unaligned sequence $\vec{x}$ for which $P(\vec{x} | \vec{\sigma}, \vec{x}_m, \vec{x}_i)$ is 1.

## Training an HPM

Our hidden Potts model design contains two independent pieces, the pHMM-like and Potts-like parts of the model, that we train separately. We train the pHMM-like elements ($P_t$, $P_i$) using the HMMER software package program `hmmbuild` with an MSA as input [5]. Additionally, `hmmbuild` annotates which columns in the input MSA correspond to consensus positions: columns with fewer than 50% gaps are deemed to be consensus. Using only the

consensus columns, we then train a Potts model, $P_{Potts}$, with the Gremlin structure prediction software [11]. Gremlin uses an approximate pseudolikelihood maximization method to estimate the $h_k$ and $e_{kl}$ Potts model terms [37]. We combine $P_t$, $P_i$, and $P_{Potts}$ with our own code (hpmbuild) to produce a fully parameterized HPM.

A Potts model produced by Gremlin is not normalized, as the partition function $Z$ is not calculated. Therefore, we are only able to calculate the probability of a given $\vec{x}_m$ under a Potts model up to $Z$.

$$P_{Potts}^* (\vec{x}_m) = ZP_{Potts} (\vec{x}_m) \tag{7}$$

Thus, we only know the marginal probability of a sequence under an HPM up to a factor of Z.

$$P^* (\vec{x}) = ZP (\vec{x}) \tag{8}$$

Implications of using unnormalized probability distributions are discussed below.

## Importance sampling alignment algorithm

In order to use HPMs in remote homology search, we need algorithms to efficiently align and score sequences of varying length with a parameterized HPM. Profile HMMs and profile SCFGs use dynamic programming algorithms to optimally align and score sequences in polynomial time. However, dynamic programming algorithms do not work in models like HPMs with non-nested correlation terms ($e_{kl}$'s).

Aligning a sequence consists of finding the best possible combination of $\vec{\sigma}$, $\vec{x}_m$, and $\vec{x}_i$ for a given sequence under an HPM.

$$\{\vec{x}_m, \vec{x}_i, \vec{\sigma}\}_{ali} = \text{argmax}_{\{\vec{x}_m, \vec{x}_i, \vec{\sigma}\}} P^* (\vec{x}, \vec{x}_m, \vec{x}_i, \vec{\sigma}) \tag{9}$$

Scoring a sequence entails summing the unnormalized joint probabilities for a given sequence $\vec{x}$ and the set of *all possible* combinations of $\vec{\sigma}$, $\vec{x}_m$, and $\vec{x}_i$ to find the unnormalized probability that the HPM generated the sequence, $P^*(\vec{x})$.

$$P^*(\vec{x}) = \sum_{\{\vec{\sigma}, \vec{x}_m, \vec{x}_i\}} P^* (\vec{x}, \vec{x}_m, \vec{x}_i, \vec{\sigma}) \tag{10}$$

Evaluating all possible combinations of $\vec{\sigma}$, $\vec{x}_m$, and $\vec{x}_i$ by brute force enumeration is computationally intractable, so we need an approximate method to align and score sequences efficiently. One could imagine using Monte Carlo integration, randomly sampling a finite number of combinations from $P^* (\vec{x}_m, \vec{x}_i, \vec{\sigma})$ and approximating the marginalization sum above to obtain $P^* (\vec{x})$. However, the space of possible combinations is enormous, and few of the sampled combinations would yield non-negligible probabilities $P^* (\vec{x}, \vec{x}_m, \vec{x}_i, \vec{\sigma})$.

To more efficiently score and align sequences, we use a related method, *importance sampling*, illustrated in Fig 2 and described in detail in S1 Appendix. Under importance sampling, instead of sampling $\vec{x}_m$, $\vec{x}_i$, and $\vec{\sigma}$ from the prior distribution $P(\vec{x}_m, \vec{x}_i, \vec{\sigma})$, we instead sample a smaller number of paths from a different distribution, the "proposal" distribution, $Q(\vec{x}_m, \vec{x}_i, \vec{\sigma})$, in which the probability mass for all three variables is much more concentrated.

Using importance sampling, for R samples, $P^* (\vec{x})$ is approximated by:

$$P^* (\vec{x}) \approx \frac{1}{R} \sum_{r=1}^{R} P (\vec{x}|\vec{x}_m^{(r)}, \vec{x}_i^{(r)}, \vec{\sigma}^{(r)}) \frac{P^* (\vec{x}_m^{(r)}, \vec{x}_i^{(r)}, \vec{\sigma}^{(r)})}{Q (\vec{x}_m^{(r)}, \vec{x}_i^{(r)}, \vec{\sigma}^{(r)})} \tag{11}$$

The approximate alignment is the sampled combination of $\vec{\sigma}$, $\vec{x}_m$, and $\vec{x}_i$ that maximizes the numerator of the importance sampling sum.

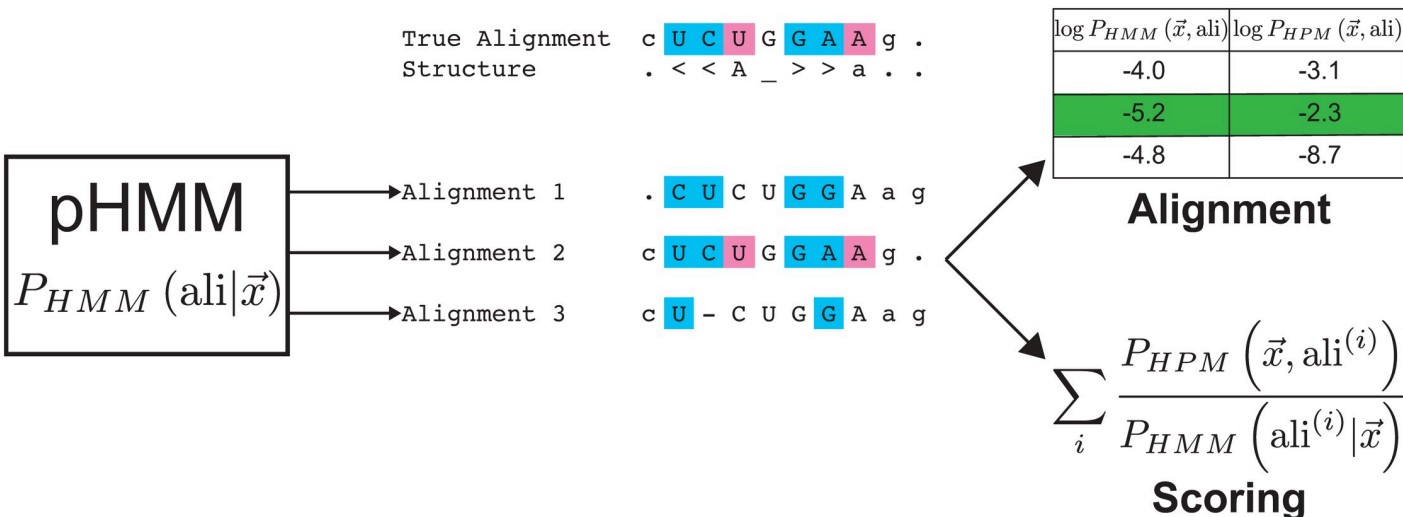

**Fig 2. Importance sampling alignment algorithm schematic.** Toy example of aligning an RNA sequence CUCUGGAAG to models of its sequence/structure consensus, where the true structural alignment has two nested base pairs (brackets in structure line) and one pseudoknot ('Aa'). Suboptimal alignments of the sequence are sampled probabilistically using a pHMM. A pHMM does not capture residue correlations due to base-pairing, so only some proposed alignments satisfy the expected consensus nested (cyan) or pseudoknotted (pink) base pairs. The proposed alignments are re-scored and re-ranked under the HPM, which does capture correlation structure; the correct alignment with the highest probability under the HPM is identified (green), and the sequence's total probability $P(\vec{x})$ is obtained by importance-weighted summation over the sampled alignments.

For a proposal distribution, we use the posterior probability of an HPM path, match sequence, and insert sequence given an unaligned sequence under a pHMM, $P_{HMM}(\vec{x}_m, \vec{x}_i, \vec{\sigma}|\vec{x})$. We show in S1 Appendix that $P_{HMM}(\vec{x}_m, \vec{x}_i, \vec{\sigma}|\vec{x})$ is equivalent to $P_{HMM}(\vec{\pi}|\vec{x})$, where $\vec{\pi}$ is a pHMM-style path with delete states. The $P_{HMM}(\vec{\pi}|\vec{x})$ posterior distribution over all possible alignments of a sequence to a pHMM can be sampled exactly and efficiently by stochastic tracebacks of the Forward dynamic programming matrix [1, 38], requiring one $\mathcal{O}(ML)$ time calculation of the Forward matrix followed by an $\mathcal{O}(L)$ time stochastic traceback per sampled alignment. (Recall $M$ is the number of match states in the HPM and $L$ is the number of non-gap characters in unaligned sequence $\vec{x}$.) Re-scoring each sampled alignment with the HPM takes $\mathcal{O}(M^2 + L)$ time, where the most expensive step is evaluating all-vs-all pairwise $e_{kl}$ terms in the Potts emission of $\vec{x}_m$. Overall, for a proposed sample of $R$ alignments, our approach takes $\mathcal{O}(ML + RL + RM^2)$ time, typically dominated by the $RM^2$ term.

Intuitively, our approach assumes that a pHMM has the alignment reasonably well constrained, though not correct in detail, which is our experience with alignments of remotely homologous sequences that fall near detection thresholds. Because the pHMM neglects residue correlations, the alignment that best satisfies conserved correlation patterns is often not the optimal pHMM alignment. So long as this alignment is probable enough to be present in a large stochastic sample from the pHMM's distribution over alignment uncertainty, rescoring that sample with the more complex HPM will find it.

We have taken $10^6$ samples from $P_{HMM}(\vec{\pi}|\vec{x})$ per sequence when aligning and scoring, which takes roughly 1 minute per sequence for typical $M$ and $L$. Our tests indicate that the importance sampling approximation for $P^*(\vec{x})$ generally converges by $10^6$ samples, and the optimal alignment is generally stable (see S1 Appendix for more information).

To score homology searches, we use a standard log odds score, in which we compare $P^* (\vec{x})$ to the probability that $\vec{x}$ was generated by a null model, $P_N (\vec{x})$ [1].

$$S^* (\vec{x}) = \log \left( \frac{P^* (\vec{x})}{P_N (\vec{x})} \right) \tag{12}$$

Ideally, we would use a normalized HPM distribution $P(\vec{x})$, which requires knowing the partition function, to produce a normalized log odds score $S(\vec{x})$. Calculating $Z$ exactly using all possible match state subsequences is intractable. As $Z$ is a constant for a particular model, log odds scores can be compared relatively across different searches with a given query HPM, but not qualitatively across different query HPMs (with different unknown $Z$'s). However, knowing $Z$ and having normalized log odds scores would not give us statistical significance values (E- and p-values), which also depend on the size of the target database. For future work, both the partition function and statistical significance will have to be evaluated empirically. We can still use the importance sampling alignment algorithm to find a maximum likelihood alignment of a target sequence to a query HPM or to calculate the posterior probabilities of alignments of target sequences to an HPM, tasks for which $Z$ cancels and statistical significance estimations are not strictly needed.

## HPMs perform promisingly

To test the performance of HPMs in remote homology search and alignment, we designed benchmark experiments using a few deep, well-curated RNA alignments. We choose to test on RNA alignments for three reasons. First, secondary and tertiary structure in RNA is largely dictated by nucleotide base-pairing, which makes the $e_{kl}$ terms in the HPM easily interpretable. Second, pSCFG methods already use RNA secondary structure conservation patterns, so we can compare the performance of HPMs not just to sequence-only models like pHMMs, but also to existing models that already capture most RNA pairwise correlation structure. Finally, because the residue correlation structure in RNAs largely consists of disjoint base pairs, we can make some back-of-the-envelope observations about how much additional information content an HPM will capture compared to a pSCFG or a pHMM. This estimation serves as a gauge of how much we might be able to improve homology search and alignment, as discussed next.

**Information content of RNA multiple sequence alignments.**   We can get a sense of how much additional statistical signal is capturable by HPMs compared to pSCFGs and pHMMs from the Shannon relative entropies (information content) of single MSA columns and column pairs. The relative entropy of a residue emission probability distribution for a single position [39] is roughly the expected score per position in a pHMM. The difference between the relative entropy of the joint emission distribution for two positions and the sum of their individual relative entropies is the *mutual information*, the extra information (in bits) gained by treating the two positions as a correlated pair. We define secondary structure information content as the summed mutual information of all nested base pairs in an RNA consensus structure; this is a measure of how much statistical signal a pSCFG gains over a pHMM for an RNA alignment. Similarly, we can define "tertiary structure" information content as the summed mutual information of all other disjoint base pairs in an RNA consensus structure, such as pseudoknots, not included in a nested pSCFG description. This is not a complete picture of the statistical signal capturable by an HPM, because the measure neglects non-disjoint pairs (e.g. base triples) and other higher-order correlations that an HPM can also capture. However, RNA pseudoknots are one of the most important features we aim to capture in more powerful sequence homology search and alignment models.

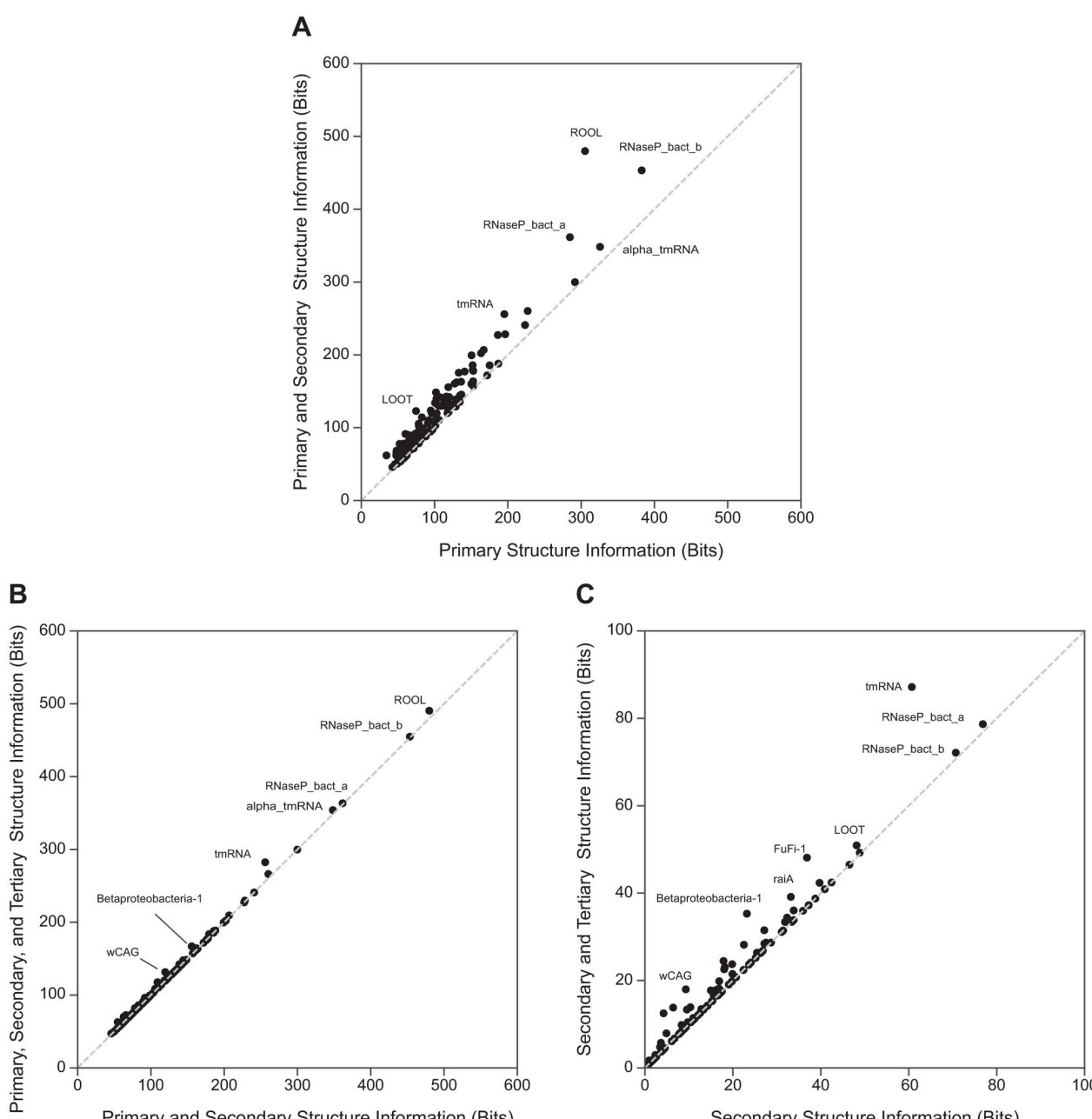

**Fig 3. Additional information contained in pairwise covariation in RNA MSAs.** For each of the 127 Rfam 14.1 seed MSAs with more than 100 sequences [36], we infer a predicted consensus structure (including nested and non-nested base pairs) using CaCoFold and R-scape [41, 42]. (A) Primary information content (X axis) versus the sum of primary and secondary information content (Y axis). (B) Sum of primary and secondary information content (X axis) versus the information content from all three levels of structure (Y axis) (C) Secondary information content (X axis) versus the sum of information content from secondary and tertiary structure (Y axis). Not included in (C) is ROOL (RF03087), with 305.3 bits of primary information, 174.5 bits of secondary information, and 10.7 bits of tertiary information.

As shown in Fig 3, most of the information content in Rfam RNA MSAs results from primary and secondary structure conservation. Tertiary structure information content contributes to a lesser degree. These results indicate that HPMs have the potential to only slightly gain sensitivity over pSCFGs in RNA homology search. However, when trying to improve the

**Table 1. Benchmark dataset statistics and training alignment information content.**

| Family | $N_{train}$ | $N_{test}$ | Consensus Length | 1˚ info (bits) | 2˚ info (bits) | 3˚ info (bits) |
|---|---|---|---|---|---|---|
| tRNA | 1357 | 30 | 72 | 40.3 | 26.7 | 1.2 |
| Twister | 1005 | 40 | 65 | 57.5 | 4.5 | 1.3 |
| SAM | 192 | 8 | 108 | 121.8 | 11.1 | 0.0 |

Hand-curated MSAs are split into training and test sets based on [45]. For each training MSA, information content in the primary sequence (in bits) is calculated [39], while information in secondary structure (nested base pairs) and tertiary structure (all other disjoint pairwise interactions between sites) is estimated using mutual information [6]. Each family's consensus structure is inferred using CaCoFold and R-scape on the training alignment [41, 42]. Though R-Scape identifies no tertiary structure using the SAM riboswitch training alignment, a four-base pair pseudoknot has been observed experimentally [46]. This lack of pseudoknot detection is a characteristic of our SAM training alignment; R-scape predicts the pseudoknot when analyzing the RF00162 seed alignment.

sensitivity of homology search, even small increases in signal are potentially useful; for instance, a 2-bit increase in log odds score corresponds to a 4-fold decrease in E-value [40].

**Benchmark procedure.** We choose to use a small number of well-studied sequence families of known 3D structure, as opposed to larger and more systematic benchmarks, in order to be able to study individual alignment results in detail. We use three RNA alignments for these experiments: an alignment of 1415 transfer RNA (tRNA) sequences [43], a 1446-sequence Twister type P1 ribozyme MSA [44], and the 433-sequence SAM riboswitch seed alignment from Rfam family RF00162 [36]. Summary statistics for each benchmark dataset are shown in Table 1. Each training alignment contains at most a small amount of information content resulting from pairwise covariation induced by conserved tertiary structure.

To create benchmark datasets, we divide each reference MSA into an in-clade training set and a remotely homologous test set, as we have done in previous work on RNA homology search [45]. To create a challenging test and simulate a search for very distantly related sequences, we cluster sequences by single linkage clustering and then split the clusters so that the following conditions are satisfied:

- No test sequence is more than 60% identical to any training sequence.

- No two test sequences are more than 70% identical.

For scoring benchmarks, we add 200,000 randomly-generated, non-homologous decoy sequences to the test set. We use synthetic sequences in order to avoid penalizing a method that identifies remote, previously unknown evolutionary relationships. Decoys are created with characters drawn i.i.d from the nucleotide composition of the positive test sequences, with the length of each decoy matching a randomly-selected positive test sequence. We score all of the test and decoy sequences with homology models built using the training MSAs. We measure scoring effectiveness by creating a ROC plot that shows true positive rate, or sensitivity, as a function of false positive rate.

To test the effectiveness of HPMs in remote homology alignment, we measure how accurately each model aligns remote homologs relative to the original alignment. We define accuracy to be the fraction of residues in consensus columns aligned correctly across all test sequences. We do not measure alignment accuracy in insert columns, as pHMMs, pSCFGs, and HPMs do not align characters in insert states.

**Scoring and alignment benchmark results.** The chief motivation in using HPMs for homology search is the potential for conserved patterns of pairwise evolution to be captured by the all-by-all pairwise $e_{kl}$ terms. Are the $e_{kl}$'s useful for homology search? To answer this question, we compare the performance of default HPMs with all $e_{kl}$'s included (red curves in Fig 4) against HPMs for which the $e_{kl}$ terms are constrained to 0 when training with Gremlin

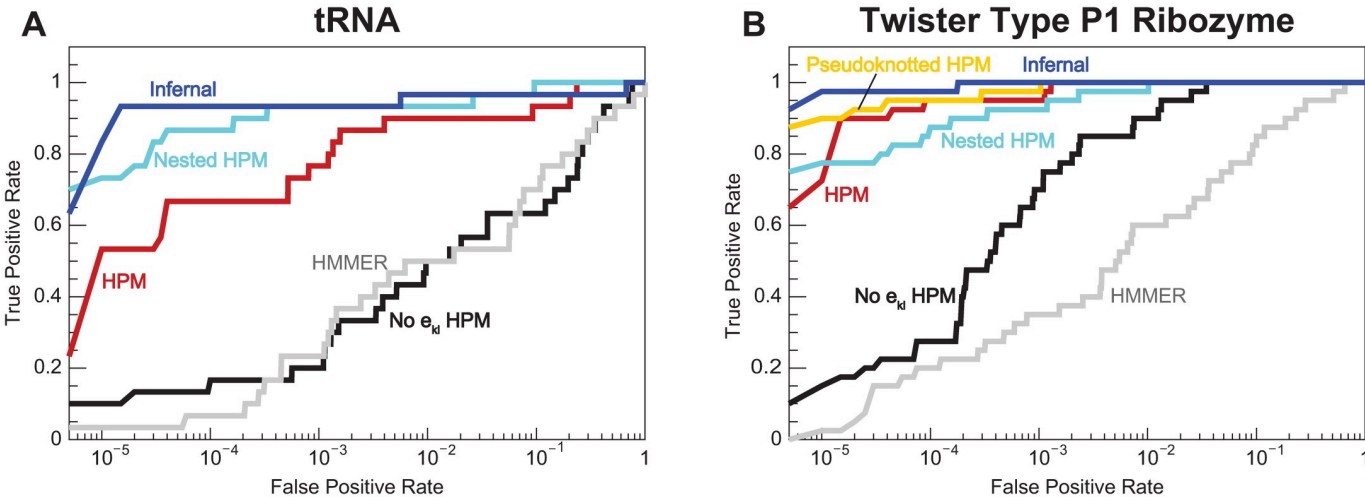

**Fig 4. Remote homology scoring benchmark results.** Receiver operating characteristic (ROC) plots for the tRNA (A) and Twister type P1 ribozyme (B) benchmarks. The X axis is the fraction of decoys that score higher than a certain threshold (false positive rate), and the Y axis is the fraction of homologous test sequences that score higher than the same threshold (true positive rate, or sensitivity). In the SAM riboswitch benchmark, each model perfectly discriminates homologs from decoys.

("no $e_{kl}$" HPMs, black curves). A no $e_{kl}$ HPM is similar to a pHMM, but there three key differences: gaps are represented by deletion characters rather than deletion states; sequences are aligned using importance sampling rather than the Viterbi algorithm; and Gremlin uses different prior information than HMMER. The HPM outperforms the no $e_{kl}$ HPM in both the tRNA and Twister ribozyme benchmarks, showing greater sensitivity in the ROC plots in Fig 4. (In the SAM riboswitch benchmark, every method tested is able to fully separate homologous test and decoy sequences.) In the three alignment benchmarks (see Table 2), the all-by-all HPM is more accurate than the no $e_{kl}$ HPM. We conclude $e_{kl}$'s generally add sensitivity to remote homology search and alignment.

But does our proof of principle HPM implementation outperform existing methods? We compare the performance of the HPM to HMMER and Infernal, pHMM and pSCFG homology search tools, respectively [7, 47] (gray and blue lines in Fig 4). The HPM generally outperforms HMMER, only falling short in the SAM riboswitch alignment benchmark. Nevertheless, Infernal is usually more sensitive than the HPM, with the only exception being the Twister ribozyme alignment benchmark.

Why is our HPM implementation not outperforming Infernal? One difference between the two methods is model parameterization. Maximum a posteriori pSCFG parameters are calculated analytically from weighted residue frequencies and priors, and Infernal's choices for weighting and priors have been optimized for homology search and alignment over decades. In contrast, pseudolikelihood maximization of Potts model parameters is an approximation, and it has not previously been used to train homology search and alignment models. To gain more insight, we bypassed pseudolikelihood maximization by creating "masked" HPMs with non-zero $e_{kl}$ terms restricted to annotated nested base pairs ("nested HPM", cyan line in Fig 4) or to all annotated disjoint base pairs ("pseudoknotted" HPM, yellow line). For a model with a correlation structure consisting solely of disjoint base pairs, maximum a posteriori Potts model parameters can be estimated directly and analytically like pSCFG parameters by setting $e_{kl}$ terms to negative log joint probabilities and $h_k$, $h_l$ terms to zero for $k$, $l$ pairs. These "masked" HPMs were almost always more sensitive than the original HPM, with performance

**Table 2. Remote homology alignment benchmark results.**

| Model | Total Accuracy | Nested Base Pairs | Pseudoknotted Base Pairs | Other Positions |
|---|---|---|---|---|
| **tRNA** | | | | |
| HPM | **0.863** | 0.871 | N/A | 0.850 |
| No $e_{kl}$ HPM | **0.700** | 0.696 | N/A | 0.706 |
| HMMER | **0.641** | 0.642 | N/A | 0.639 |
| Infernal | **0.901** | 0.938 | N/A | 0.843 |
| Nested HPM | **0.879** | 0.900 | N/A | 0.846 |
| **Twister Ribozyme** | | | | |
| HPM | **0.762** | 0.762 | 0.940 | 0.669 |
| No $e_{kl}$ HPM | **0.622** | 0.592 | 0.848 | 0.550 |
| HMMER | **0.683** | 0.675 | 0.873 | 0.597 |
| Infernal | **0.718** | 0.766 | 0.880 | 0.563 |
| Nested HPM | **0.784** | 0.782 | 0.980 | 0.686 |
| Pseudoknotted HPM | **0.794** | 0.797 | 0.980 | 0.694 |
| **SAM Riboswitch** | | | | |
| HPM | **0.748** | 0.753 | 0.918 | 0.711 |
| No $e_{kl}$ HPM | **0.733** | 0.737 | 0.918 | 0.693 |
| HMMER | **0.847** | 0.851 | 0.984 | 0.817 |
| Infernal | **0.977** | 0.993 | 0.984 | 0.956 |
| Nested HPM | **0.911** | 0.930 | 0.984 | 0.873 |
| Pseudoknotted HPM | **0.894** | 0.916 | 0.918 | 0.861 |

Total alignment accuracy is measured as the fraction of residues in consensus columns aligned correctly across all test sequences, relative to the reference alignment. Accuracy across consensus columns with different types of secondary structure annotation is also included.

closer to Infernal's. This result suggests using the pseudolikelihood approximation to train a Potts model is one of the obstacles to using HPMs for homology search.

The improved performance of the masked HPMs relative to the default HPM prompted us to look further into the HPM's Potts model parameterization. We examine pairwise probabilities of nucleotides under the HPM at annotated base pairs, estimated using Markov chain Monte Carlo. These probabilities do not reflect the observed pairwise frequencies in the training MSA. For instance, Fig 5 shows disagreement between HPM probabilities and training MSA frequencies for an annotated base pair in the SAM riboswitch training MSA. We see many such examples in the SAM riboswitch example.

However, the Gremlin-trained Potts model is accurate in its intended purpose: predicting molecular structure from aligned sequence data. For all three benchmarks, Gremlin's predicted contacts heavily overlap with annotated base pairs or other observed structural interactions (Fig 6).

**Characterizing HPM parameterization issues with simulated data.** As an additional check on our results and parameterization issues, we performed positive controls on synthetic data, using a synthetic multiple sequence alignment sampled from an HPM with known parameters. This procedure ensures both the HPM parameters and the sequence alignment are ground truth. To execute these experiments with realistic parameters (as opposed to random), we use a multi-step procedure starting from a real input MSA, the Twister Ribozyme benchmark training MSA (here called "MSA1"). We train an HPM on MSA1 to obtain the "training HPM" with ground truth parameters. We generate 1000 independent aligned sequences from the training HPM by Markov chain Monte Carlo (MCMC); this alignment is

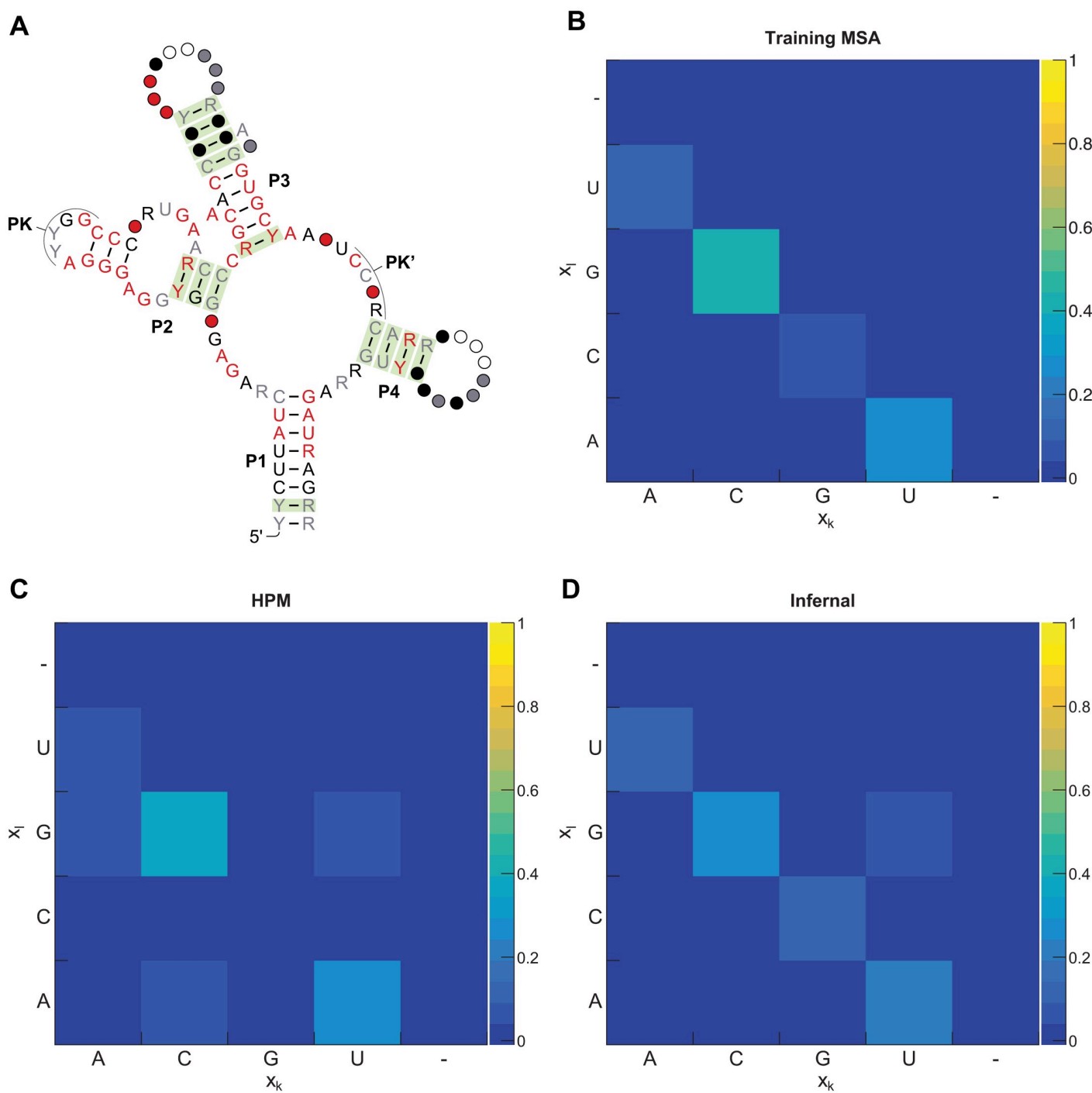

**Fig 5. Hidden Potts model emission probabilities do not match training alignment statistics.** (A) Rfam consensus structure for the class I SAM riboswitch. Base pairs supported by statistically significant covariation in analysis by R-scape in the RF00162 seed alignment are shaded green [41]. (B) Observed pairwise nucleotide frequencies for one base pair in the P3 stem (sites 52 and 62) in our RF00162 benchmark training alignment (192 sequences). (C) Pairwise nucleotide probabilities at sites 52 and 62 under the RF00162 training HPM. (D) Pairwise nucleotide probabilities at sites 52 and 62 under the RF00162 training Infernal pSCFG. Infernal's informative priors lead to a UG base pair being given significant probability despite being rarely seen in the training data. However, as U is much more common than G at site 52 and vice versa at site 62, Infernal assigns a higher probably to UG than GU at these sites.

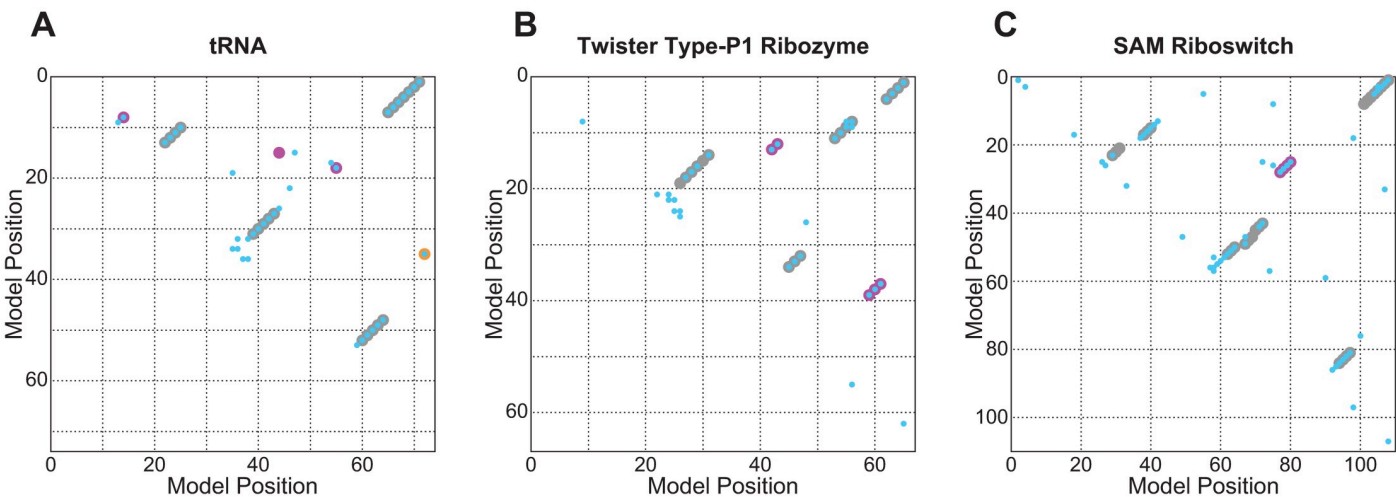

**Fig 6. Gremlin-trained Potts models accurately predict 3D RNA contacts.** Top $\frac{M}{2}$ predicted 3D contacts from Gremlin-trained Potts models (cyan) against annotated nested base pairs (gray) and annotated tertiary contacts/pseudoknotted positions (pink) from the tRNA (A), Twister type P1 ribozyme (B), and SAM riboswitch (C) training alignments. For tRNA, annotated tertiary contacts are from a yeast tRNA-phe crystal structure [48]. A non-structural covariation between the tRNA discriminator base and the middle nucleotide in the anticodon (caused by aminoacyl-tRNA synthetase recognition preferences, not conserved base-pairing) is noted in orange [49].

our ground truth MSA ("MSA2"). Now we train a new HPM on MSA2 (the "synthetic HPM"). We use this experiment to test MCMC convergence, alignment accuracy, and parameter fitting. Aligning MSA2 sequences to the synthetic HPM assesses alignment accuracy. Beyond comparing raw Potts model parameters, we can also assess how well HPM parameterization matches marginal single-site residue probabilities, marginal pairwise residue probabilities, and mutual information (pairwise correlation). We perform this comparison by generating a third alignment of 1000 sequences ("MSA3") from the synthetic HPM and comparing MSA2 versus MSA3 statistics.

This positive control yields mixed results. We observe adequate convergence of our MCMC sampling of sequences (Fig 7A) and HPM alignment accuracy under ideal synthetic conditions (alignment accuracy of 0.975, compared to 0.972 with Infernal). However, the most striking result from these experiments reveals how pseudolikelihood parameterization tends to suppress and underestimate pairwise correlations. The synthetic HPM's Potts parameters are flattened relative to the training HPM, especially the pairwise $e_{kl}$ terms (Fig 7B and 7C). Although marginal single-site and pairwise residue probabilities are reproduced reasonably accurately (Fig 7D and 7E), pairwise correlation (mutual information) is systematically underestimated (Fig 7F). We initially found this result surprising and counterintuitive, since mutual information is calculated from the single-site and pairwise probabilities. However, we find that the errors in estimating marginal single-site and pairwise probabilities are correlated, not independent, and in the direction of suppressing pairwise correlation.

Since the goal of using HPMs is to capture pairwise correlation structure in sequence alignments, the failure of pseudolikelihood maximization training to accurately reproduce pairwise mutual information is a major issue for remote homology search, and it may explain why HPMs are not performing better than Infernal in our benchmarks. Others have previously shown that training Potts models with pseudolikelihood maximization does not accurately reproduce training alignment statistics even when estimating 3D contacts accurately [50, 51]. In a way, we find this result encouraging; as we discuss below, there are alternative training methods to the commonly used pseudolikelihood maximization approach.

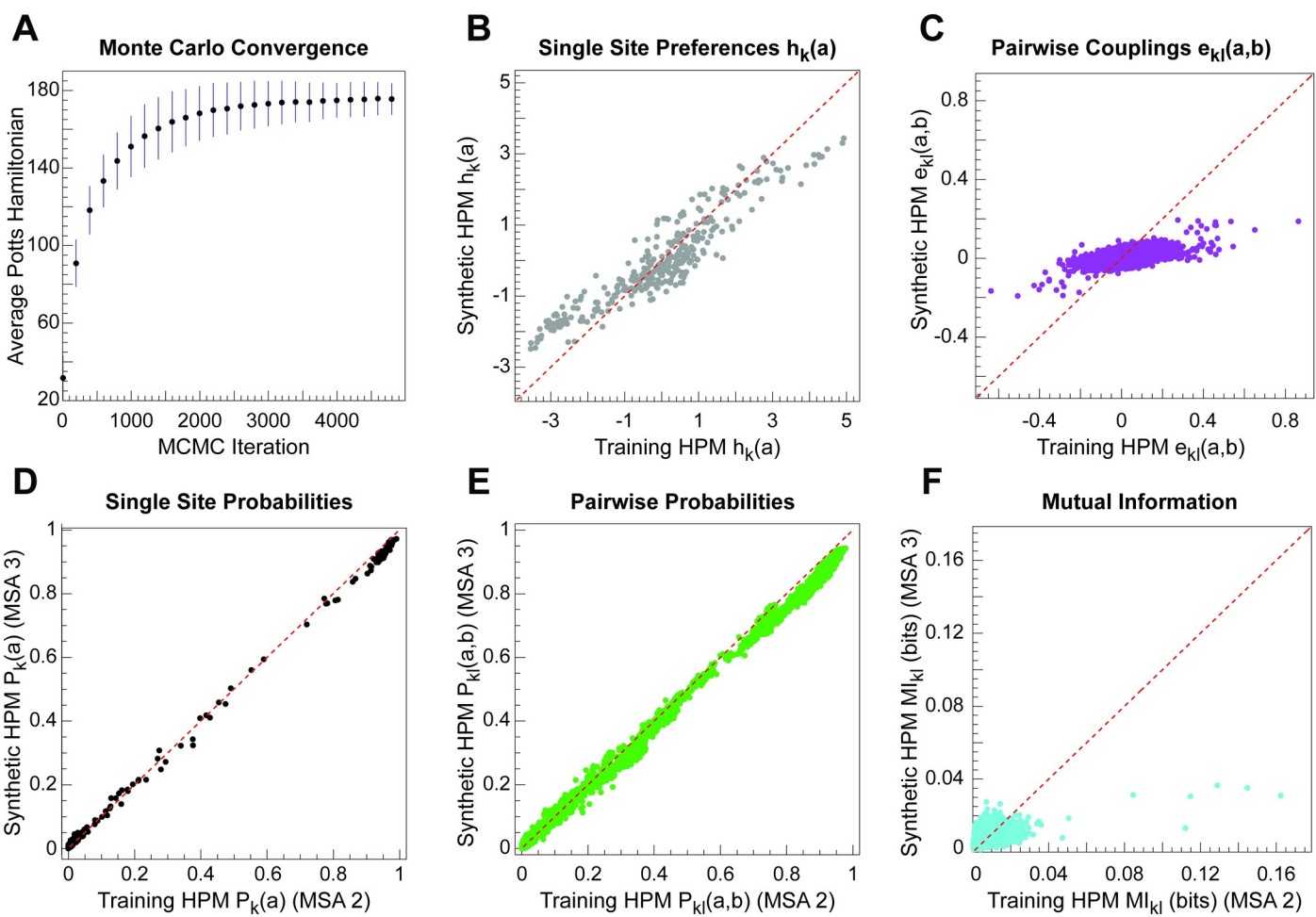

**Fig 7. Pseudolikelihood maximization does not fully recapitulate the pairwise correlation structure of input alignments.** (A) Average value of the Potts Hamiltonian at intermediate iterations of the Markov chain Monte Carlo sequence emission algorithm across 1000 synthetic sequences. Error bars represent standard deviation. We accept sequences after 5000 iterations. (B-C) Comparison of single-site (A) and pairwise (B) Potts model terms of a ground truth HPM ("training HPM") versus in an HPM trained on aligned sequences emitted from the ground truth HPM ("synthetic HPM"). (D-F) Comparisons of marginal single-site probabilities (D), marginal pairwise probabilities (E), and mutual information (F) for the posterior sequence distributions sampled from the training HPM (statistics estimated from MSA2) against the synthetic HPM (statistics estimated from MSA3).

## Discussion

We present hidden Potts models, generative probability models for sequence homology search and alignment. An HPM is a hybrid between a Potts model, successfully used in molecular structure prediction, and a profile HMM, used in protein and nucleic acid homology search and alignment. The hybrid model configuration uses advantages inherent to both models. By using a pHMM's transition architecture, an HPM can model variable length sequences, a feature previously incompatible with Potts models. The all-by-all pairwise $e_{kl}$ terms in an HPM's emission process, as in a Potts model, allow for non-nested structural interactions like pseudo-knots to be captured, going beyond the primary-sequence capability of a pHMM and the nested base pair modeling of a pSCFG. Finally, we have developed an efficient algorithm that uses importance sampling to align and score variable length sequences to an HPM.

In initial benchmarking of RNA remote homology search and alignment, we found that HPMs perform promisingly. In experiments where we compare HPMs to HPMs with pairwise

$e_{kl}$ terms forced to zero in training (thus a level comparison, within the same HPM implementation, of using versus not using correlation information), fully parameterized HPMs produce more accurate alignments and are generally more sensitive for homology searches. HPMs also generally outperformed HMMER, an existing pHMM method. This indicates that hidden Potts models capture useful conserved higher-order correlation structure information in an alignment-capable model.

On the other hand, our pilot implementation of HPMs generally underperforms Infernal, an existing pSCFG implementation for capturing nested pairwise correlations in conserved RNA secondary structure. This suggests to us that the gains from being able to model pseudo-knots and other non-nested RNA correlations are outweighed by deficiencies in our pilot implementation. Our additional experiments implicate model parameterization by pseudolikelihood maximization as a key issue. When we bypass pseudolikelihood maximization by using "masked" HPMs limited to nonzero maximum likelihood $e_{kl}$'s only for a disjoint set of base pairs, HPM performance tends much closer to Infernal performance. However, there are other Potts model training methods besides pseudolikeihood maximization, as reviewed in [51]. Additionally, we are not yet using any informative prior distributions in the Potts parameterization, comparable to the use of informative mixture Dirichlet priors in pHMMs and pSCFGs.

Besides improved parameterization, there are other issues to address to make HPMs the basis for useful homology search and alignment software tools. First, while the importance sampling alignment algorithm is efficient relative to directly enumerating over all possible alignments, it still takes seconds to minutes to align a single sequence. The method will need to be greatly accelerated. Second, while HPMs are able to handle insertions relative to a consensus with a standard affine gap-open, gap-extend probability, they do not explicitly model deletions. It would be desirable to find a cleaner model of deletions. Other work has attempted to combine Potts models with insertions and deletions using techniques based in statistical physics, though this method has not yet been applied to remote homology search and alignment [52]. Third, we only do "glocal" alignment, where a complete match to the HPM consensus model is identified in a possibly longer target sequence. We do not yet see how to do local sequence alignment to an HPM.

Future work could better characterize the performance of Potts model based methods in remote homology search. In [53], Haldane and Levy study how training alignment depth affects Potts model protein structure and fitness prediction, but no work has studied how the number of sequences affects sensitivity in remote homology search with RNA. In addition, while we use synthetic decoys with characters drawn i.i.d. to benchmark our proof-of-principle HPM implementation for simplicity, we have previously benchmarked more mature homology search methods with decoys that more closely resemble real biological sequences [47]. Further work could use synthetic decoys generated in a different manner by shuffling real biological sequences.

Another recent paper uses a different method to align protein and RNA sequences to Potts models [54]. DCAlign, developed by Muntoni et al., uses an algorithm based on message passing rather than importance sampling. They report promising results when comparing alignment accuracy to HMMER and Infernal. However, they do not attempt to score remotely homologous sequences.

Although we chose to do our initial testing with RNA sequence alignments, protein sequence homology search and alignment will also be of interest in future work. Unlike the case for pSCFGs for RNA, the state of the art for protein sequence homology search and alignment remains primary sequence methods such as HMMER and BLAST. Evolutionarily conserved protein structure creates a complex correlation structure unlike the simpler pairwise correlation patterns that dominate RNA analysis; pairwise correlations in protein alignments

are difficult to treat by anything simpler than an all-by-all network in which *any* pair of sites is potentially correlated, making protein sequence analysis ripe for Potts models.

## Materials and methods

### HPM software implementation

Our prototype HPM software implementation builds hidden Potts models from multiple sequence alignments, aligns and scores sequences with an HPM using importance sampling, and generates sequences from an HPM using Markov chain Monte Carlo. The software uses functions from HMMER version 4 (in development, see https://github.com/EddyRivasLab/hmmer/tree/h4-develop) and the Easel sequence library (see https://github.com/EddyRivasLab/easel).

To train Potts models, we use a modified version of the Gremlin C++ software package [11], included with our HPM implementation. Code is available at https://github.com/gwwilburn/WilburnEddy20_HPM. A tarball of our software is also included in S1 Code.

### HPM training

HPMs are trained by our program hpmbuild. $P_t$ transition parameters and $P_i$ insert emission parameters are taken from a pHMM trained by hmmbuild (HMMER v4). $P_{Potts}$ emission parameters are taken from a model trained by Gremlin C++ (version 1.0) modified to use Henikoff position-based sequence weights in the training procedure [55], the same relative weighting scheme used by HMMER.

### Masked HPM training

So-called masked HPMs, with only a disjoint subset of $e_{kl}$ terms allowed to be non-zero, are trained by our program hpmbuild_masked. Annotated base pairs are determined by the secondary structure consensus line in an input Stockholm MSA file. Potts model parameters are negative log likelihoods estimated from weighted counts with a + 1 Laplace pseudocount. Henikoff position-based sequence weights are used [55]. For sites belonging to annotated base pairs, $e_{kl}$'s are trained using the weighted pairwise nucleotide frequencies in the input MSA; these sites' $h_k$ terms set to zero. For unpaired sites, $h_k$ terms are trained using the weighted single-site nucleotide frequencies, and the corresponding $e_{kl}$'s are set to zero.

### Aligning and scoring sequences with an HPM

Sequences are aligned to and scored by an HPM with our program hpmscoreIS. This program takes an HPM, a pHMM with the same number of match states, and a sequence file as input. It returns the unnormalized log odds scores for each sequence estimated using importance sampling with the pHMM as a proposal distribution. Additionally, aligned sequences are outputted to a Stockholm MSA file. For each input sequence, 1 million alignments are sampled from the pHMM.

### Estimation of single-site and pairwise probabilities under an HPM

Probabilities are estimated empirically from 10,000 synthetic sequences emitted from an HPM using our program hpmemit.

We use Markov chain Monte Carlo to emit match sequence $\vec{x}_m$ from the Potts model. Each sequence is generated independently using the Metropolis-Hastings algorithm with a burn-in period of 5000 steps.

To create the HPM path $\vec{\sigma}$, we perform a random walk through the HPM's state transitions. Given $\vec{\sigma}$, we emit insertion sequence $\vec{x}_i$ by drawing residues independently from $P_i(\vec{x}_i|\vec{\sigma})$.

### Experiment with simulated sequences

1000 synthetic sequences are emitted from the Twister ribozyme benchmark training HPM using our program `hpmemit`. Each sequence is generated independently using the Metropolis-Hastings algorithm with a burn-in period of 5000 steps.

An HPM is then trained on the resulting synthetic alignment using `hmmbuild`, Gremlin, and `hpmbuild`. This model's single-site and pairwise frequencies are estimated empirically as before using sequences generated by `hpmemit`.

The other homology models are then trained on the synthetic MSA. When training with HMMER and Infernal, entropy weighting is turned off using `hmmbuild --enone` and `cmbuild --enone`, respectively. As we emit sequences from a glocal HPM, we aslo build the CM such that the flanking insert transitions are learned using the `--iflank` option.

For alignment accuracy tests, the synthetic sequences are then "realigned" to the models trained on the synthetic MSA.

Step-by-step instructions for recreating the results with synthetic sequences are included with S1 Code.

### Software and database versions used

For building pHMMs, we use HMMER 4 (in development). The code is available at https://github.com/EddyRivasLab/hmmer, inside the branch `h4-develop`. The specific version of code we use is archived under tag 7994cc7 at https://github.com/EddyRivasLab/hmmer/commits/h4-develop. The software is also included in S1 Code.

We use HMMER 4 functions to create programs `hmmalign_uniglocal` and `hmmscore_uniglocal` that glocally score and align sequences with pHMMs, respectively. These programs are included in the HPM prototype software implementation.

For analysis with pSCFGs, we use Infernal version 1.1.3 [7]. The code is available at http://eddylab.org/infernal/ and included with S1 Code.

Rfam version 14.1 is used for the SAM riboswitch anecdote [36].

For consensus RNA structure prediction by comparative sequence analysis, we used CaCo-Fold in version 1.4.0 of the R-scape software package [41, 42].

### Benchmark datasets

Training and test alignments used for each anecdote are included as supplementary material with S1 Code. Benchmark procedures follow methods in [45].

### Analysis of alignments

RNA multiple sequence alignments were visualized using the Ralee RNA alignment editor in Emacs [56].

## Supporting information

**S1 Appendix. Derivation of importance sampling alignment algorithm (PDF).**
(PDF)

**S1 Code. Contains the HPM software implementation, versions of HMMER and Infernal used in benchmarking, the Easel sequence library, and benchmark datasets (ZIP).**
(GZ)

## Acknowledgments

We thank Elena Rivas, Tom Jones, Nick Carter, and Sergey Ovchinnikov for insightful discussions throughout the project. Additionally, we thank Eric Nawrocki and members of the Eddy and Rivas labs for comments on the manuscript. Some ideas in this work were conceived at workshops hosted at the Centro de Ciencias de Benasque Pedro Pascual (Benasque, Spain), and the Aspen Center for Physics. Computations were run on the Cannon cluster, supported by the Harvard FAS Division of Science's Research Computing Group.

## Author Contributions

**Conceptualization:** Grey W. Wilburn, Sean R. Eddy.

**Data curation:** Grey W. Wilburn.

**Formal analysis:** Grey W. Wilburn.

**Funding acquisition:** Sean R. Eddy.

**Investigation:** Grey W. Wilburn.

**Methodology:** Grey W. Wilburn, Sean R. Eddy.

**Project administration:** Sean R. Eddy.

**Resources:** Sean R. Eddy.

**Software:** Grey W. Wilburn.

**Supervision:** Sean R. Eddy.

**Validation:** Grey W. Wilburn.

**Visualization:** Grey W. Wilburn.

**Writing – original draft:** Grey W. Wilburn.

**Writing – review & editing:** Grey W. Wilburn, Sean R. Eddy.

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
