## [Decision Letter · Decision Letter 0]

18 Aug 2020

Dear Dr. Eddy,

Thank you very much for submitting your manuscript "Remote homology search with hidden Potts models" for consideration at PLOS Computational Biology.

As with all papers reviewed by the journal, your manuscript was reviewed by members of the editorial board and by several independent reviewers. In light of the reviews (below this email), we would like to invite the resubmission of a revised version that takes into account the reviewers' comments.

In particular, please consider Reviewer 1's suggestion for improving clarity of the paper, Reviewer 3's suggestion for providing results with simulations and provide additional user documentation to make the code easily usable and results reproducible.

We cannot make any decision about publication until we have seen the revised manuscript and your response to the reviewers' comments. Your revised manuscript is also likely to be sent to reviewers for further evaluation.

Sincerely,

Sushmita Roy, Ph.D.

Associate Editor

PLOS Computational Biology

Jason Papin

Editor-in-Chief

PLOS Computational Biology

Reviewer's Responses to Questions

**Comments to the Authors:**

Reviewer #1: Recent work has shown that deep multiple-sequence alignments can be analyzed to find correlations between columns that indicate interactions between residues (nucleotides or amino acids). Much of this work has used "Potts models". The paper under review considers the use of Potts models to perform homology search, i.e., to find sequences that are likely to be homologous to sequences in an existing multiple-sequence alignment. The paper concludes (1) that the extra information in the tertiary interactions that Potts models consider shows promise for improving homology searches, but (2) the approximation in the Potts model leads to inferior discrimination compared to a model analogous to "Covariance Models", which are currently the best general method for RNA homology search. Therefore, it's probably necessary (in future work) to formulate a probabilistic model more like a covariance model, rather than a Potts model. Additionally, the paper devises a method, based on importance sampling, to estimate the likelihood that a given sequence is generated by a given Potts model, a problem whose exact solution is almost certainly intractable.

Homology search is a fundamental and important task. Improvements in accuracy would positively impact many research questions. Moreover, the paper's basic idea (that tertiary interactions could improve accuracy) is very reasonable. I found the paper generally well executed, and its conclusions convince me. There are two aspects that could be regarded as weaknesses, but I think are okay. First, the experiments use only 3 different RNAs, making them arguably somewhat anecdotal. However, I can't credibly argue that the overall conclusions (e.g. in the first paragraph of my review) are likely to change with more RNAs. Therefore, I can't ask the authors to produce more data -- the paper's conclusions are already convincing (at least, to me). The second potential weakness is that the paper is arguably a kind of negative result, in that a main conclusion is that HPM don't work that well, and some other (related) model is likely needed in order to usefully exploit information in interactions. However, I think the basic approach is a reasonable one that was deserving of investigation, and the paper gives important information on what kind of approaches are likely to work better.

My remaining comments are quite minor, mostly directed at improving the descriptions in a few details.

MINOR COMMENTS

W.r.t. this sentence: "A Potts model expresses the probability of a homologous sequence as a function of primary conservation and all possible pairwise correlations between all consensus sites in a biological sequence (i.e., consensus columns in a multiple sequence alignment)." : for readers not in the field, it might be good to define the concept of consensus columns explicitly. It's kind of hinted at, but never defined.

The journal for ref 50 is missing. I see that this is a preprint, but there needs to be more information on where it is.

I found the notation P_m(x_m) a bit strange (e.g., in equation 2). I think this is the probability of x_m according to the model 'm'. Wouldn't it be more conventional to write this as P(x_m|M), and state that M is the model? Also, I don't get why the model is broken into sub-models (as I understand it), e.g., p_t(\\rho).

In equations (5) and (6), lower-case 'p' occurs in the right-hand side of the equation. I assume these should be a capital 'P', e.g. the probability of x_i_j in equation (5)? Otherwise, I'm not clear on what they represent.

In equation (6), the symbol \\rho_n represents the states of the HPM. This could be explicitly stated. Secondly, I think it might help if the states were represented by s_n. In this equation the p and \\rho look similar. Also, I got to this point in the paper believing that the path variable was \\overrightarrow{p} and not \\overrightarrow{\\rho}. I think s would be more conventional.

I'm not sure I understand the description of \\Lambda in equation (6). Does \\rho include deletion characters or not? I think it does (since deletion characters must correspond to states), but the phrasing makes me wonder. Also, why is the variable L introduced here, when it's only actually used a few pages later? I found the latter clause in the sentence (page 7, line 165) more confusing than helpful. If it's left in, I'd recommend changing the semicolon to a new sentence (it looked like a comma when I first read it), and something like "Here, \\Lambda is the total number of states in \\rho. Given that \\rho includes deletion characters, \\Lambda is at least as large as the number of non-gap characters in x." But really, I think this should be moved later in the text.

It would be nice to remind the reader of the meaning of M on page 10? Also, I think the variable L should be defined here. I believe this is the same as the L on page 7.

Since Potts models don't have a notion of insertions, the Potts model is presumably using every column in the input alignment (even gappy ones), right? In this case, is HMMER told to also treat all columns (even gappy ones) as consensus columns? I think this would be good to clarify.

I have some questions about this sentence: "As Z is a constant for a particular model, scores can be compared relatively within a single database search with a given query HPM, but not qualitatively across different query HPMs (with different unknown Z's).". I get why searches with different HPM models (which have different implicit values of Z) can't be compared. But, this sentence implies that searches with the same query HPM of different databases are not comparable (because it specifies "within a single database"). Is this simply because the sizes of the databases are (in general) different, and so the statistical significance of a given score changes? Or is there some other reason?

I agree with the following statement, but I think it needs more support. Not necessarily objective data, but something like a hypothetical scenario, or some kind of an argument illustrating the intuition. "However, when trying to improve the sensitivity of homology search, even small increases in signal are potentially useful."

In Fig. 4B, the black line is an HPM with e_ij set to zero. Isn't this equivalent to a pHMM? Why does it perform so much better than HMMER in this test?

I suspect a bit more analysis of Fig. 4 would be helpful for some readers, and to some extent for me. Mainly, it's apparent that Infernal (Fig 4D) allows for a significant probability for G-U base pairs in comparison to the training MSA (Fig 4B). This is presumably the result of priors that lead Infernal to anticipate the Watson-Crick-compatible G-U pair. Thus, Infernal is distorting the input distribution, but in a desired way. However, I myself am a bit puzzled as to why it doesn't also allow for U-G base pairs. At any rate, the G-A and A-C pairs that the HPM allows (Fig 4C) are not desired. I think a brief explanation of these issues in Fig. 4 would help.

Minor grammar issues / typos:

"the the training alignment"

"with performance is closer to Infernal's"

"Besides improved parameterization, there are other issues to address to make HPMs as the basis for useful homology search and alignment software tools." : delete "as"

"grand number n/a"  "grant number N/A"

SUGGESTION OUT OF THE SCOPE OF PAPER

If HPMs prove successful, couldn't they be added as a new final step in Infernal's search pipeline? Perhaps the time that HPM alignments take would not longer be such a significant problem in this context. I think this is out of the scope of the paper, but the paper could mention this possible if the authors wish. (If not, I'm fine with that; no need to rebut.)

Reviewer #2: Reproducibility report has been uploaded as an attachment.

Reviewer #3: The submission introduces "hidden Potts models", a novel approach to identify remote homology which extends the profile-HMM approach to Potts models used to model co-evolving sites for structural prediction.

The problem tackled is central to computational biology. The new method is imaginative and thought-provoking. The manuscript is very clearly written.

I am generally very positive about the manuscript. I believe a better characterisation of the approach under the model through simulation and better code usability would facilitate follow-up work on this promising new line of research.

Major points

-----------------

* Given the novelty of the approach and implementation, I would expect some validation using simulated data under the model, to convince that the approach works as expected. For instance, I would think that the following experiments would be informative. e.g.

- parameter estimation: generate sequences under models with specific parameters, and demonstrate that the parameters can be estimated correctly from the data.

- alignment accuracy: can the method recover the correct alignment (i.e. residue-level homology) among a bunch of sequences generated by a model.

- convergence of monte carlo sampling: how long is sampling needed to obtain satisfactory results in the above experiments?

Minor points

-----------------

* "We use synthetic sequences in order to avoid penalizing a method that identifies remote, previously unknown evolutionary relationships. Decoys are created with characters drawn i.i.d from the nucleotide composition of the positive test sequences, with the length of each decoy matching a randomly-selected positive test sequence." One risk with this approach is that the generated sequences have properties that are pretty different from real sequences (i.i.d.). I feel this should be at least discussed. But two simple strategies which could be of interest would be to invert the sequences (which keeps local composition) or shuffling the sequence in local windows.

* I would expect the "no e_{kl} HPM" model to performs similarly or worse than pHMM. Yet it gave better performance on the tRNA model. Any sense why this could be the case?

* "In the three alignment benchmarks (see table 2), the all-by-all HPM is more accurate than the no e_{kl} HPM. We conclude e_{kl}’s generally add sensitivity to remote homology search and alignment." Were parameters also fitted with the constraint that e_{kl} = 0?

* Potts model fitting need very large MSAs. This is likely to be the case here. It would be interesting to see if the number of sequences used for training has an impact on the performance. But in any case this could be discussed.

* Code is provided, but this is quite some way to being "usable". There is hardly any documentation, and no test. The code has some external dependencies which makes compiling non-trivial. No binary is provided. I understand the authors regard their contribution more of a proof of concept than a tool (and indeed the senior author has an outstanding track record providing usable tools) but a little effort toward making the code more usable would surely encourage reuse and extensions. (Actually, I later found that the code and readme files provided as supplementary materials provides sample files and compiling instructions—please include these on the GitHub repo as well please, as most readers are likely to retrieve the code from Github rather than a supplementary file to the manuscript).

**Have all data underlying the figures and results presented in the manuscript been provided?**

Reviewer #1: Yes

Reviewer #2: None

Reviewer #3: Yes

PLOS authors have the option to publish the peer review history of their article (what does this mean?). If published, this will include your full peer review and any attached files.

Reviewer #1: No

Reviewer #2: **Yes: **Anand K. Rampadarath

Reviewer #3: **Yes: **Christophe Dessimoz
---

## [Decision Letter · Decision Letter 1]

27 Oct 2020

Dear Dr. Eddy,

We are pleased to inform you that your manuscript 'Remote homology search with hidden Potts models' has been provisionally accepted for publication in PLOS Computational Biology.

Best regards,

Sushmita Roy, Ph.D.

Associate Editor

PLOS Computational Biology

Jason Papin

Editor-in-Chief

PLOS Computational Biology

Reviewer's Responses to Questions

**Comments to the Authors:**

Reviewer #1: The authors have addressed all of my comments. I think the paper can be published in its present form.

Reviewer #2: Reproducibility report has been uploaded as an attachment.

Reviewer #3: I thank the authors for addressing my points thoroughly, and in particular for performing the simulation-based analyses I requested. I have no further reservation.

**Have all data underlying the figures and results presented in the manuscript been provided?**

Reviewer #1: Yes

Reviewer #2: None

Reviewer #3: Yes

PLOS authors have the option to publish the peer review history of their article (what does this mean?). If published, this will include your full peer review and any attached files.

Reviewer #1: No

Reviewer #2: **Yes: **Anand K. Rampadarath

Reviewer #3: **Yes: **Christophe Dessimoz

---

## [Editor Report · Acceptance letter]

25 Nov 2020

PCOMPBIOL-D-20-01035R1 

Remote homology search with hidden Potts models

Dear Dr Eddy,

I am pleased to inform you that your manuscript has been formally accepted for publication in PLOS Computational Biology. Your manuscript is now with our production department and you will be notified of the publication date in due course.

With kind regards,

Nicola Davies
